

# Exposure to Floods, Climate Change, and Poverty in Vietnam

Mook Bangalore[1,2], Andrew Smith[3], Ted Veldkamp[4]

[1] Grantham Research Institute and Department of Geography and Environment, London School of Economics, London, UK, WC2A 2AE

[2] World Bank, Washington, DC, USA

[3] SSBN Ltd, Bristol, UK

[4] Institute for Environmental Studies, Vrije Universiteit, Amsterdam

*Correspondence to: Mook Bangalore (M.Bangalore@lse.ac.uk)*





**Abstract.** With 70 percent of its population living in coastal areas and low-lying deltas, Vietnam is highly exposed to riverine and coastal flooding. This paper examines the exposure of the population and poor people in particular to current and future flooding in Vietnam and specifically in Ho Chi Minh City, using new high-resolution flood hazard maps and spatial socioeconomic data. The national-level analysis finds that a third of today's population is already exposed to a flood, which occurs once every 25 years, assuming no protection. For the same return period flood under current socioeconomic conditions, climate change may increase the number exposed to 38 to 46 percent of the population. Climate change impacts can make frequent events as important as rare ones: the estimates suggest a 25-year flood under future conditions can expose more people than a 200-year flood under current conditions. Although poor districts are not found to be more exposed to floods at the national level, the city-level analysis of Ho Chi Minh City provides evidence that slum areas are highly exposed. The results of this paper show the benefits of investing today in flood risk management, and can provide guidance as to where future investments may be targeted.

**JEL codes:** Q54, I30, Q50
**Keywords:** Floods, Poverty, Vietnam, Exposure, Urban Development



## 1. Introduction

Vietnam is a rapidly developing country highly exposed to natural hazards. One of the major natural risks the country faces is riverine and coastal flooding, due to its topography and socioeconomic concentration: Vietnam's coastline is 3,200 kilometers long and 70 percent of its population lives in coastal areas and low-lying deltas (GFDRR 2015).

Furthermore, climate change is expected to increase sea levels and the frequency and intensity of floods, globally and in Southeast Asia (IPCC 2014; World Bank 2014). Given the country's concentration of population and economic assets in exposed areas, Vietnam has been ranked among the five countries most affected by climate change: a 1 meter rise in sea level would partially inundate 11 percent of the population and 7 percent of agricultural land (World Bank and GFDRR 2011; GFDRR 2015).

Even though climate change impacts are expected to primarily occur in the future, flooding already causes major problems in Vietnam, with some segments of the population more vulnerable than others (Adger 1999; World Bank 2010; World Bank and Australian AID 2014). In particular, evidence suggests poor people are more vulnerable than the rest of the population to natural disasters such as floods, as their incomes are more dependent on weather, their housing and assets are less protected, and they are more prone to health impacts (Hallegatte et al. 2016, Chapter 3).

Poor people also have a lower capacity to cope with and adapt to shocks due to lower access to savings, borrowing, or social protection; and climate change is likely to worsen these trends (Hallegatte et al. 2016, Chapter 5).

Therefore, it is important to quantify how many people are exposed to floods, how this distribution of exposure falls upon regions and socioeconomic groups, and how climate change may influence these trends. Employing flood hazard maps and spatial socioeconomic data, this paper examines these questions in the context of Vietnam:

1. How many people are exposed currently? How might this change under climate change?
    2. Where is exposure highest currently? How might this change under climate change?
    3. How many poor people are exposed currently? How might this change under climate change?

Furthermore, given that the dynamics of poverty and natural disasters (and particularly, floods) occur at the local level, analyses at the national scale (or even at the province or district level) may miss important mechanisms and small-

scale differences, from one city block to the next. To complement the country-level analysis, we also focus at the local level within Ho Chi Minh City (HCMC), a city with high flood exposure. Here, we combine high-resolution flood hazard data with spatial data on slum location, urban expansion, and migration, to examine the distribution of exposure across poor and non-poor locations. While many studies have examined flood risk in Vietnam, many have only focused on hazard mapping. The contribution of this paper is to include the socioeconomic dimensions and examine

how flood exposure is distributed across poor and non-poor locations, at the country and city levels.

The national-level analysis finds that a third (33%) of today's population is already exposed to a 25 year event (an event with a probability of occurrence of 0.04), assuming no protection. For the same return period flood under current socioeconomic conditions, climate change may expose 38-46% of the population, depending on the severity of sea level rise. Climate change impacts may make frequent events as important as rare ones in terms of exposure: for



instance, a 25-year flood under future climate conditions exposes more people than a 200-year flood under current conditions. While poor districts are not found to be more exposed to floods at the national level, the city-level analysis of HCMC provides evidence that 68-85 percent of slum areas are exposed to floods, a higher percentage than the rest of the city. In addition to showing the benefits of investing today in flood protection, this paper provides policy implications for the design of flood risk management strategies in Vietnam.

## 2.   Literature review

In the last 30 years, floods worldwide have killed more than 500,000 people and resulted in economic losses of more than US$500 billion (Kocornik-Mina et al. 2015). It is therefore no surprise that a number of studies have examined the population and economic assets exposed to flood risk. At the global level, it is well documented that an increasing share of the population and economic assets lie in areas exposed to riverine and coastal flood risk today, and these trends show no sign of slowing down (UN-ISDR 2015; Ceola, Laio, and Montanari 2014; Jongman et al. 2014). To compound these socioeconomic changes, climate change is expected to intensify many hazards and further increase exposure: the number of people exposed to river floods could increase by 4-15% in 2030 and 12-29% in 2080 (Winsemius et al. 2015).

But only a handful of global studies have examined how this distribution of flood exposure differs between rich and poor. Kim (2012) assesses these dynamics at the country-level, and finds that poor countries tend to be more exposed to natural disasters, including floods, compared to rich countries. More recently, (Winsemius et al. 2015) examined whether poor people *within countries* are more exposed to flood risk, and found that this was the case for 60% of the 52 countries sampled.

Within Vietnam, studies suggest that floods significantly impact poverty, both quantitatively at the national level using household survey data (Bui et al. 2014) and qualitatively through focus group interviews at the local level in Ho Chi Minh City (World Bank and Australian AID 2014). One study within Vietnam examines the exposure of poor and non-poor people to floods and found that a disproportionate number of poor people live in highly-flooded areas of the Mekong Delta (Nguyen 2011).

At a more local scale and especially within cities, land and housing markets often push poorer people to settle in riskier areas. Where markets factor in hazard risks, housing is cheaper where risk is higher (Husby and Hofkes 2015). And, because poorer people have fewer financial resources to spend on housing and a generally lower willingness and ability to pay for safety, they are more likely to live in at-risk areas (Lall and Deichmann 2012; Fay 2005; Hallegatte et al. 2016).

Empirically, this higher exposure to flood risk for poor urban dwellers is found in about 75% of the countries examined by (Winsemius et al. 2015), and also when using high-resolution data on household location and flood hazards in Mumbai, India (Patankar 2015). This high exposure of the urban poor to floods has severe implications on the health of children and economic outcomes of adults, as evidenced in HCMC (World Bank and Australian AID 2014).



This paper provides an in-depth case study of floods, poverty, and climate change in Vietnam and Ho Chi Minh City, examining the exposure of the total population, and poor people in particular to current and future flood risk. It makes two contributions; the first is that it combines state-of-the-art hazard maps with socioeconomic data to examine distributional impacts of floods at the national-level in Vietnam. Most previous analyses of floods and climate change

in Vietnam at the national-level have focused on hazard mapping and not its distributional impacts (Institute of Strategy and Policy on Natural Resources and Environment 2009; Ministry of Natural Resources and Environment 2009). The second contribution is the paper's analysis of flood exposure and poverty at national and local levels: most previous analyses have focused on one or the other (Winsemius et al. 2015; World Bank and Australian AID 2014).

### 3. Data

To examine population and poverty-specific exposure to floods, we employ spatial data defining flood hazard and a number of socioeconomic characteristics representing poverty and population density.

### 3.1. Flood hazard data

### 3.1.1. Flood hazard maps for Vietnam developed for this study

For this study, we developed flood hazard maps representing riverine, flash-flood and coastal flood risk for Vietnam.

These flood hazard maps estimate the inundation depth at a grid cell level of 3 arc-seconds, (~ 90m) and provide coastal surge hazard layers, along with pluvial and fluvial layers. The maps provide information on the extent and depth of flood hazard for a specific location. For the coastal component, we explicitly model four return periods – 25, 50, 100, and 200 year events, under current and future climate conditions.

There is a significant amount of uncertainty with regards to how much sea level will rise. For that reason we model

three future climate scenarios per return period: a low, medium, and high scenario (Table 1), using estimates from the IPCC (IPCC 2014; IPCC 2007). For the fluvial and pluvial hazards, future climate scenarios were not explicitly simulated owing to the complexity and considerable uncertainties that arise (Smith et al. 2014).[1]

Although robust modeling of the magnitude of future extreme rainfall is not yet possible, heavy rainfall is expected to increase in a warmer climate, owing to the increased water holding capacity of the atmosphere. Therefore instead of

a direct modeling approach, future climate scenarios were inferred by taking flood hazard maps derived under current climate conditions for different return periods, and using them as a proxy for future climate scenarios. The return

---

[1] These uncertainties largely arise from climate models; global climate models (GCMs) struggle to represent the physical processes that produce extreme rainfall. Indeed even in higher resolution regional climate models (RCMs), heavy rainfall events are poorly represented. As a result the modelled rainfall data must be 'corrected', in order to render it realistic. The fact that the underlying models themselves cannot represent flood driving rainfall means that there is little confidence in the projections that they produce. Moreover, at the national scale there is very little river gauge data available in Vietnam. Therefore rainfall-runoff models, required to transform rainfall projections into river discharge values, would be largely un-calibrated. This adds an additional source of significant modeling uncertainty to the model cascade. The combination of poorly represented extreme rainfall in climate models, coupled with uncalibrated rainfall-runoff models, would largely render any projections of future flood risk impractical, owing to the significant uncertainties that arise.





period hazard maps used for each of the future scenarios are outlined in Table 2. Although simplistic, this method allows areas that may be impacted by increasing riverine and extreme rainfall driven flooding to be identified. Clearly there are some significant assumptions and uncertainties arising from this method. However, given the impracticalities of modeling future flood risk in Vietnam, this approach provides a plausible and practical attempt to estimate changing

flood risk at the national scale.

For each of the four return periods, four scenarios are modeled (historical, future with low sea level rise, future with medium sea level rise, and future with high sea level rise), combining the coastal and fluvial/pluvial hazard layers (Table 2). Importantly, the flood hazard models do not include flood protection (such as dikes and drainage systems), which can make a large difference in the flood hazard particularly in well-protected areas. In these well-protected

areas, our flood maps may over-estimate the flood hazard. For full details on the methodology used to produce these hazard maps, see Appendix 1.

*Table 1. Future scenarios used for Vietnam coastal modeling. RCP stands for Representative Concentration Pathway. We use two RCPs from the recent Intergovernmental Panel on Climate Change (IPCC) report (IPCC 2014) to represent a low climate change and a high climate change scenario. RCP2.6 is a low scenario consistent with*

*temperature increases of 2°C, while RCP8.5 is a high scenario consistent with temperature increases of 4°C. The A1B scenario was taken from a previous IPCC report (IPCC 2007) and represents a medium climate change scenario, in between RCP2.6 and RCP8.5.*

| Simulations | Scenario | Percentile | SLR -2100 (m) |
|---|---|---|---|
| Low | RCP 2.6 | 0.5 | 0.28 |
| Medium | A1B | 0.05 | 0.6 |
| High | RCP 8.5 | 0.95 | 0.98 |

*Table 2. Hazard map scenarios for which the modeling was conducted for Vietnam*

| Scenario | Coastal | Fluvial/Pluvial |
|---|---|---|
| 1 in 25 | 1 in 25 | 1 in 25 |
| 1 in 25 Future – Low | 1 in 25 + 28cm | 1 in 50 |
| 1 in 25 Future – Medium | 1 in 25 + 6cm | 1 in 75 |
| 1 in 25 Future – High | 1 in 25 + 98cm | 1 in 100 |
| | | |
| 1 in 50 | 1 in 50 | 1 in 50 |
| 1 in 50 Future – Low | 1 in 50 + 28cm | 1 in 75 |
| 1 in 50 Future – Medium | 1 in 50 + 6cm | 1 in 100 |
| 1 in 50 Future – High | 1 in 50 + 98cm | 1 in 200 |
| | | |
| 1 in 100 | 1 in 100 | 1 in 100 |
| 1 in 100 Future – Low | 1 in 100 + 28cm | 1 in 200 |
| 1 in 100 Future – Medium | 1 in 100 + 6cm | 1 in 250 |
| 1 in 100 Future – High | 1 in 100 + 98cm | 1 in 500 |
| | | |
| 1 in 200 | 1 in 200 | 1 in 200 |
| 1 in 200 Future – Low | 1 in 200 + 28cm | 1 in 250 |
| 1 in 200 Future – Medium | 1 in 200 + 6cm | 1 in 500 |
| 1 in 200 Future – High | 1 in 200 + 98cm | 1 in 1000 |



For most of the analyses, the "combined" maps are used, which include both coastal and the fluvial/pluvial floods. For instance, the combined maps for the 25-year return period flood (under current conditions, and low, medium, and high future conditions) are presented in Map 1. A Google Earth image of Ho Chi Minh City with the flood map for a 25-year return period with high climate change is presented in Map 2.

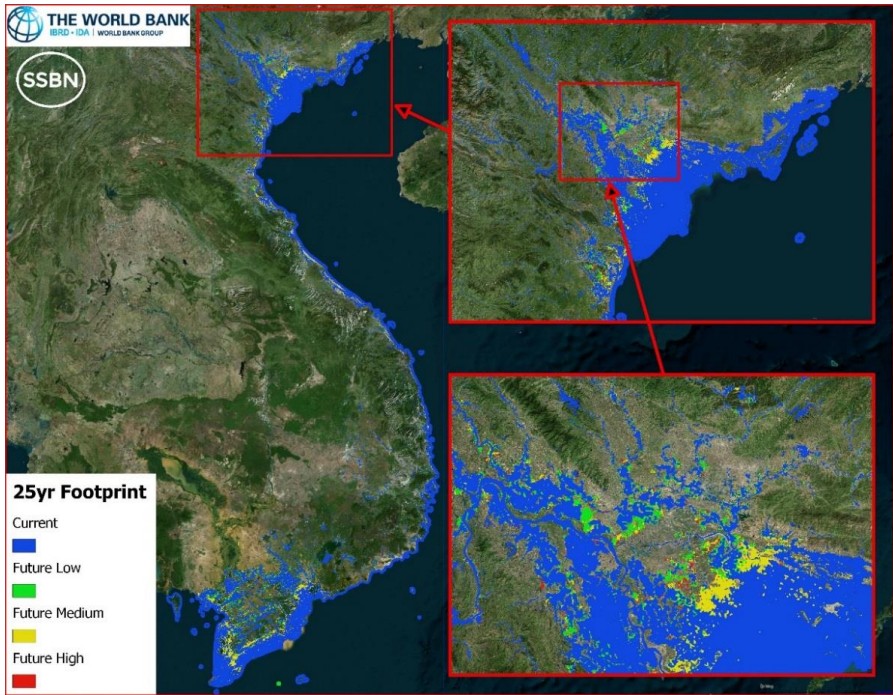

*Map 1. A visual of what the combined hazard maps (which include coastal and fluvial/pluvial) look like. The map presented here is the worse-case scenario we simulate, a 200-year return period flood with high sea level rise.*

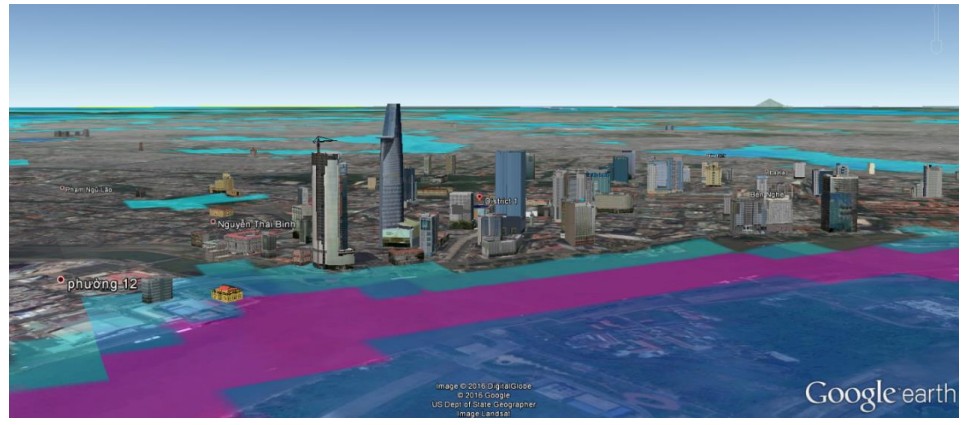

*Map 2. Google Earth image of the flood maps overlaid with the built environment in Ho Chi Minh City, for the 25-year return period under high climate change.*





### 3.1.2.    Local flood hazard maps for Ho Chi Minh City

In addition to the flood hazard maps developed for this study as described above, we use an additional set of maps produced specifically for HCMC.

The inundation maps were used in an earlier flood risk study of HCMC (Lasage et al. 2014), and were composed with
the MIKE 11 hydraulic modeling software (DHI 2003). The flood hazard maps, which have a spatial resolution of 20 meters, represent the current conditions for five return periods: 10, 25, 50, 100, and 1000 years. Future conditions, again using the five return periods, include a sea level rise scenario of +30 centimeters in the year 2050 (consistent with the "low" sea level rise used for the maps produced for this study) in combination with current river discharge (FIM 2013). Potential peaks in precipitation events and/or river discharges due to climate change are not covered by
this data set. The inundation layers for a 10, 25, and 50-year return period under current climate conditions and given a sea level rise scenario of +30 centimeters are shown in Map 3.

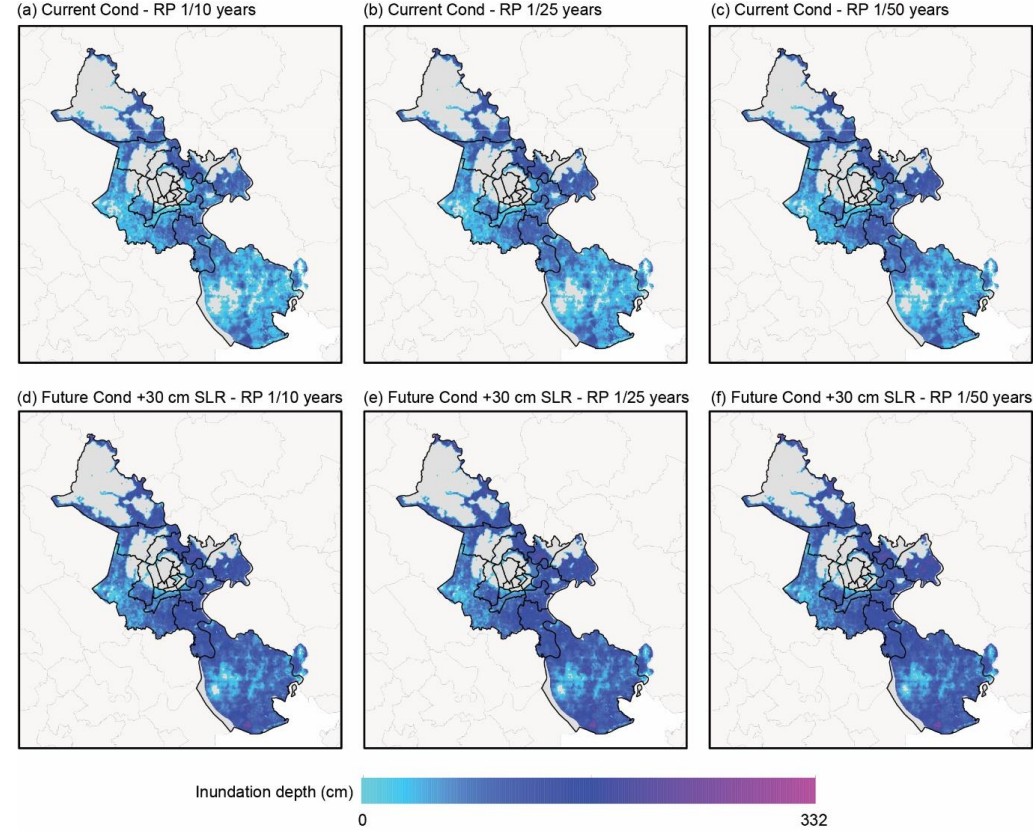

*Map 3. Flood maps showing inundation depth (cm) in case of a: (a) 10-year return period flood under current conditions, (b) 25-year return period flood under current conditions; (c) 50-year return period flood under current*
*conditions; (d) 10-year return period flood given a 30 cm sea level rise; (e) 25-year return period flood given a 30 cm sea level rise; and (f) 50-year return period flood given a 30 cm sea level rise.*




### 3.2. Socioeconomic data

#### 3.2.1.    District-level poverty and population data

At the national-level analysis, we overlay the flood hazard maps developed for this study with spatial socioeconomic data. For Vietnam, the World Bank has produced estimates of the number of people within each district who live

below the poverty line: this "poverty map" is displayed in Map 4a, and the full methodology can be found in (Lanjouw, Marra, and Nguyen 2013). In addition, we use gridded population density data with a 1km resolution from Landscan (Geographic Information Science and Technology 2015). This "population map" is displayed in Map 4b.

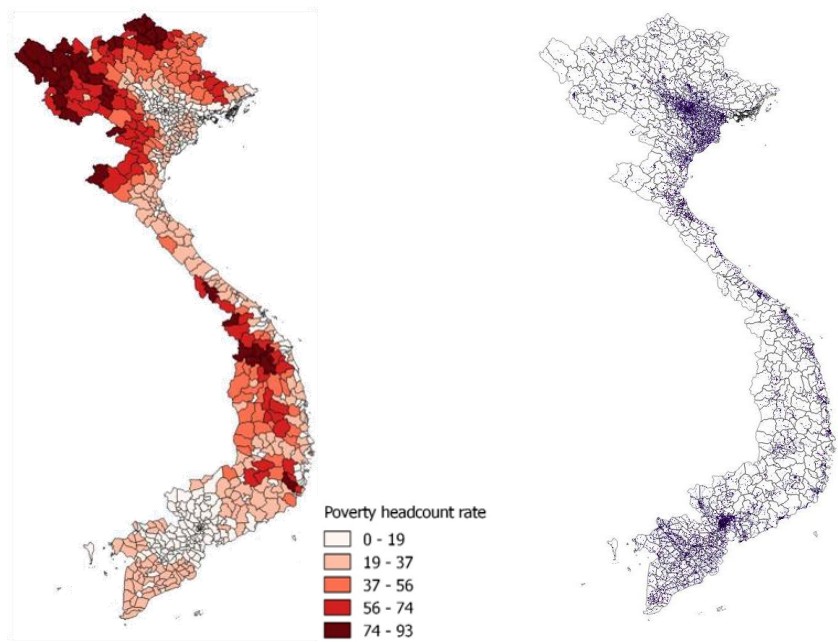

*Map 4. (a) Poverty map and (b) population density map for Vietnam at the district level. Sources: (Lanjouw, Marra,*
*and Nguyen 2013; Geographic Information Science and Technology 2015)*

#### 3.2.2.    Slum data and urban expansion data in Ho Chi Minh City

The spatial socioeconomic data set used for Ho Chi Minh City is a data set of potential slum areas and of urban expansion from 2000 to 2010, from the Platform for Urban Management and Analysis (PUMA), a city-level data set developed by the World Bank (World Bank 2015).This data was collected via satellite in the year 2012, through a

combination of visual interpretation of various sources and vintages of imagery.

To guide the identification of slums, previous work has provided information on the appearance and geographical extent of slums in HCMC. Surveys of poverty in the city find the appearance of slums in HCMC to be characterized as densely built small households and shelters that have predominantly semi-permanent character (Habitat for Humanity 2008). In terms of geographic extent, many slums are located in certain districts ( districts 2, 3, 4, 6, 8, 11,

12, Binh Thanh, Go Vap, Tan Phu) and along the Saigon River (e.g. Kenh Te, Rach Ben Nghe, Thi Nghe-Nhieu Loc





Canal, Kenh Doi, Thi Nghe Canal, Lo Gom, and Canala) (Horsley 2004; De Lay 2011; Habitat for Humanity 2008). Taking into account these spatial and geographic characteristics, the PUMA data set interprets Google Earth imagery to produce two layers of potential slum areas (PUMA 2013): areas with defined borders (polygon-data) and potential slum areas without (point-data) defined borders. In the latter case, we applied a circular buffer of 50 meters around

each point indicating a potential slum location. Evidence suggests that slum areas exist in the northern districts of HCMC (Habitat for Humanity 2008), which are not reflected in Map 5. For this reason, we ran the analyses for two samples – all the districts in the province, and only the districts with potential slums from PUMA.

PUMA also collects data on land-use change, based on satellite interpretation of land use in 2000 and 2010. The data set identifies areas of urban expansion, defined as "the extension of artificial services and associated areas". (PUMA

2013). The slum locations and locations of urban expansion in HCMC are presented in Map 5.

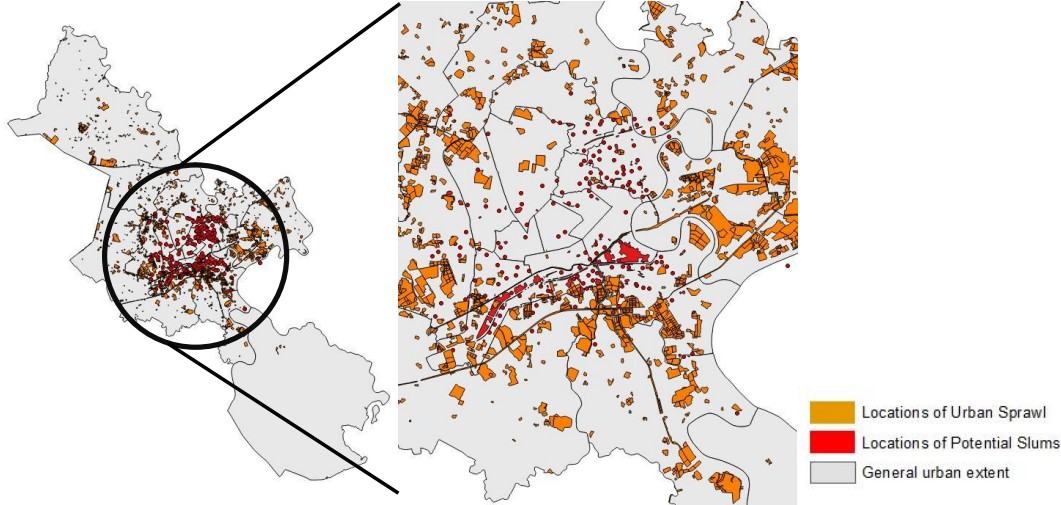

*Map 5. Location of slum areas and locations with urban expansion in the city of HCMC. Source: (PUMA 2013).*

## 4.  Methodology

### 4.1. Exposure to flooding at the national level

At the national level, we estimate per district the number of people exposed to each scenario of flooding, and the number of poor people exposed. In the flood data, we define exposed areas as those grid cells where the flood level is greater than 0; non-exposed areas are those grid cells where the flood level is zero. This is a measure of *extent* rather than *depth*, and has been used in previous studies to examine exposure to floods (Jongman et al. 2014; Winsemius et al. 2015; Ceola, Laio, and Montanari 2014). Furthermore, while we lose information by using *extent* rather than *depth*



(we have depths in our flood data), we decided to use extent since our flood data assumes no protection. Protection is more likely to impact the depth, rather than the extent, of the flood results.[2]

We then overlay this flood layer with the population density data set, to estimate the number of people per population grid cell that are exposed to floods. As the population density data set is at a lower resolution (1km) than the flood data (90m), we estimate the percentage of the population grid cell which is flooded, and multiply this percentage by the population in that grid cell. For instance, if a population grid cell has 500 people, and 10% of that cell is flooded (based on the flood data), then we estimate 50 people to be exposed to floods in that cell. In doing so, we assume that the population is evenly distributed within a grid cell.

We run this analysis for all the scenarios presented in Table 2, and aggregate our results at the district level to estimate the number of people affected. To include the poverty dimension, we use the poverty headcount rate in each district to estimate the percentage of poor people exposed. For instance, if 20,000 people are exposed to floods in District X, and District X has a poverty headcount rate of 20%, 1,000 poor people are exposed to floods in that district. In this analysis, we assume that poverty is evenly distributed within a district.

**4.2. Slum and urban expansion exposure in Ho Chi Minh City**

For the HCMC analysis, we estimate the general exposure to flooding, for the whole province of HCMC and in each of its 24 districts. The flood maps used here are based on a model of HCMC, and are not the same map as used in the figurative example in Section 4.1.

Exposure to flooding was again evaluated using flood extent (we also evaluate flood depth, for full results, see Appendix 2). We examine the flood extent in three areas: for all urban areas (the whole HCMC province), for those areas defined as potential slums (from the PUMA data set), to examine how exposure to floods is different in slum areas. We do the same for areas defined as urban expansion locations (also from the PUMA data set) to evaluate whether new urban developments within the province of HCMC take place in flood prone areas.

Again we use a number of events, from the case of regular flooding (10-year event) to more extreme flooding events (1000-year event). Moreover, we examine how this exposure changes due to climate change (proxied by sea level rise changes), by running the analysis with flood hazard maps taking into account a 30 cm sea level rise. In each district and across the whole city, we examine the percentage of area within each of the three categories (all urban areas, slums, and urban expansion areas) that is exposed to floods (that is, where flood depth > 0cm) and the percentage which is not exposed to floods (that is, where flood depth = 0cm). We then compare these values across the three categories.

**5. Results**

---

[2] There is also a good reason for examining extent over depth, in terms of the hazard modeling; flood depths within a large scale flood model are very uncertain, and there is much more certainty about extents.



### 5.1. National-level analysis for poverty and exposure to floods

#### 5.1.1.   Flood risks (with and without climate change)

For the entire country of Vietnam, at the district level, we estimate the total number of people and the share of the population who are exposed to floods. In the results presented, we examine the four scenarios for the 25-year, 50-year, 100-year and 200-year return period flood – a historical scenario, and three scenarios representing future climate: a low, medium, and high scenarios.

We aggregate the results at the country level.[3] A third (33%) of today's population is already exposed to a 25-year flood in Vietnam, assuming no protection (such as dikes and drainage systems), which can make a large difference in the flood hazard particularly in well-protected areas. In these well-protected areas, our flood maps may over-estimate the flood hazard.

When including climate change, this percentage increases by 13-27%, depending on the severity of sea level rise. This increase in exposure is due to the concentration of the population in coastal areas. For the 50-year flood, more than a third (38%) of today's population is already exposed. Given climate change, this number is expected to increase by 7-21% (resulting in overall exposure of between 40 and 48%) for the same return period (50-year). For a 100- and 200-year flood under a high climate scenario, more than half of the population is exposed.

Climate change impacts can be seen in these exposure numbers – for instance, a 50-year flood with medium climate change impacts has the same exposure of a 200-year historical flood (at 44%), while almost half the country's population (48%) is exposed to a 50-year flood with high climate impacts. Full results are presented in Table 3.

*Table 3. Population exposed to flood risk in Vietnam, across the 16 flood hazard scenarios examined.*

| Scenario | Exposure | Return period | | | |
|---|---|---|---|---|---|
| | | 25 | 50 | 100 | 200 |
| Historical | Estimated population exposed (m) | 30.17 | 34.30 | 38.35 | 40.43 |
| | Percentage of today's population | 33% | 38% | 42% | 44% |
| Low climate change | Estimated population exposed (m) | 34.78 | 36.87 | 40.91 | 42.32 |
| | Percentage of today's population | 38% | 40% | 45% | 46% |
| | Increase due to climate change | 13% | 7% | 6% | 4% |
| Medium climate change | Estimated population exposed (m) | 38.03 | 40.22 | 43.34 | 45.16 |
| | Percentage of today's population | 42% | 44% | 48% | 50% |
| | Increase due to climate change | 21% | 15% | 11% | 10% |
| High climate change | Estimated population exposed (m) | 41.46 | 43.36 | 46.13 | 48.72 |
| | Percentage of today's population | 46% | 48% | 51% | 53% |
| | Increase due to climate change | 27% | 21% | 17% | 17% |

[3] Results presented are similar to a previous study analyzing the exposure to a 100-year return period flood without climate change impacts, which finds 40 million people to be exposed to that event (Jongman et al. 2014).





But these national results on exposure are not evenly be distributed across the country. The spatial analysis also allows us to examine which districts have the highest absolute and the highest relative exposure. We present results for the 25-year flood, for a historical and a high climate scenario (results on geographical extent for other scenarios are similar). For absolute exposure, the largest number of people exposed are found in the Mekong Delta, the Red River Delta, and the Southeast Coast (Map 6 and Map 7). But the relative exposure (that is, the % of the district population which is exposed to floods) shows a larger spread. Most areas in the country – including the North Central Coast and the Northeast – have high percentages of their populations residing in flood-prone areas (Map 9).

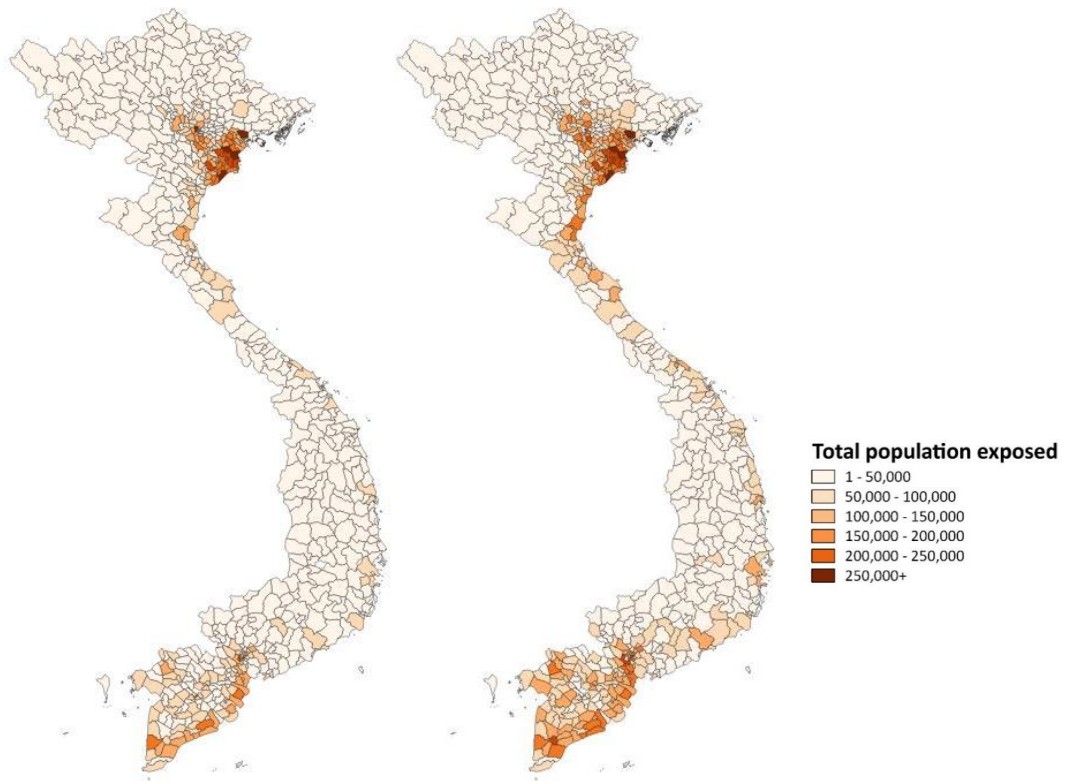

*Map 6. Absolute exposure at the district level (total number of people in a district exposed), for a 25-year historical flood (left) and a 25-year historical flood under high climate change (right).*





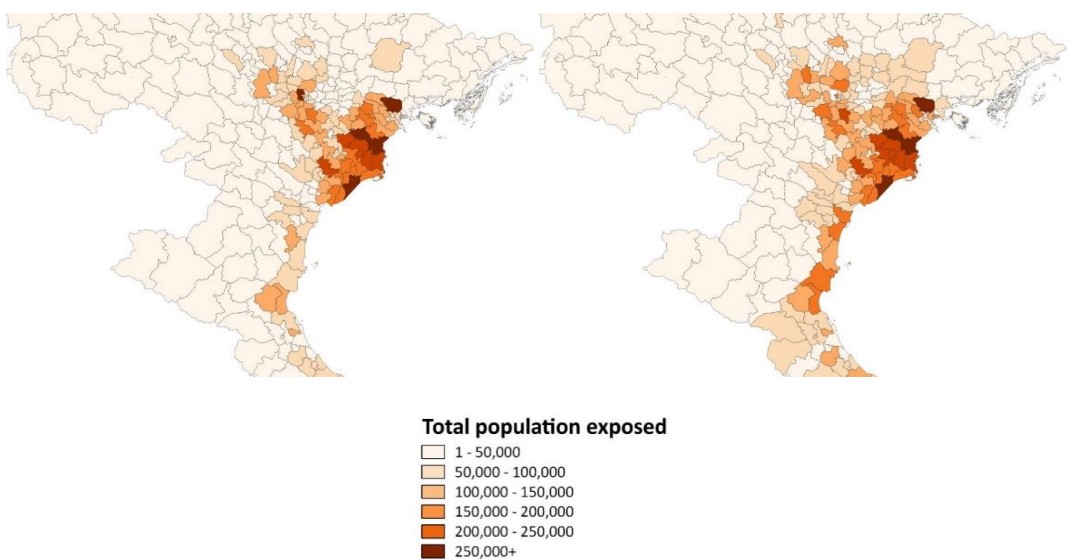

*Map 7. Total population exposed in the Red River Delta for historical 25-year flood (left) and 25-year flood with high climate impacts (right)*

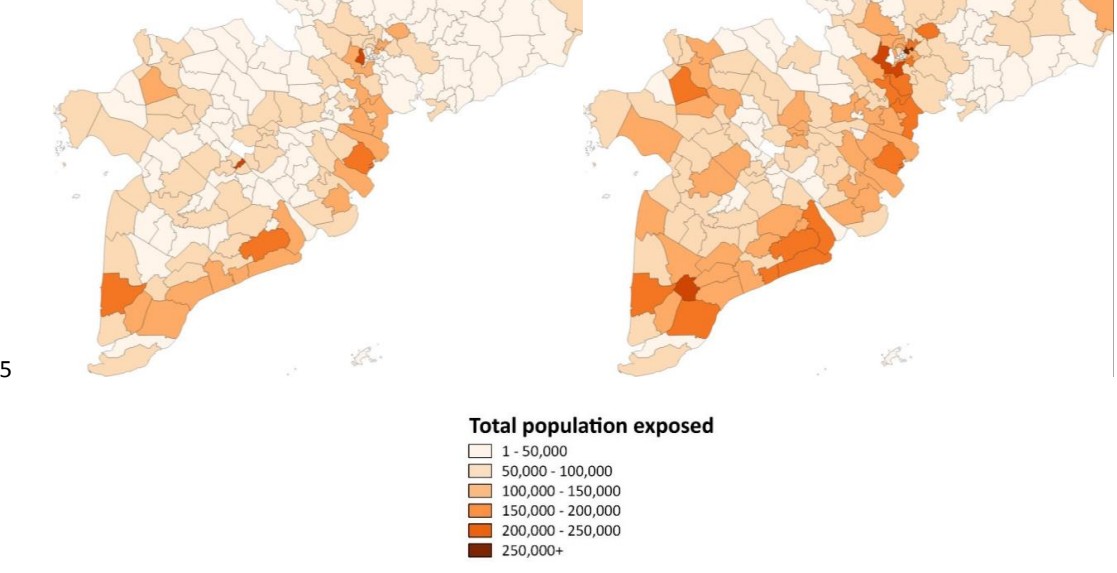

*Map 8. Total population exposed in the Mekong for historical 25-year flood (left) and 25-year flood with high climate impacts (right)*





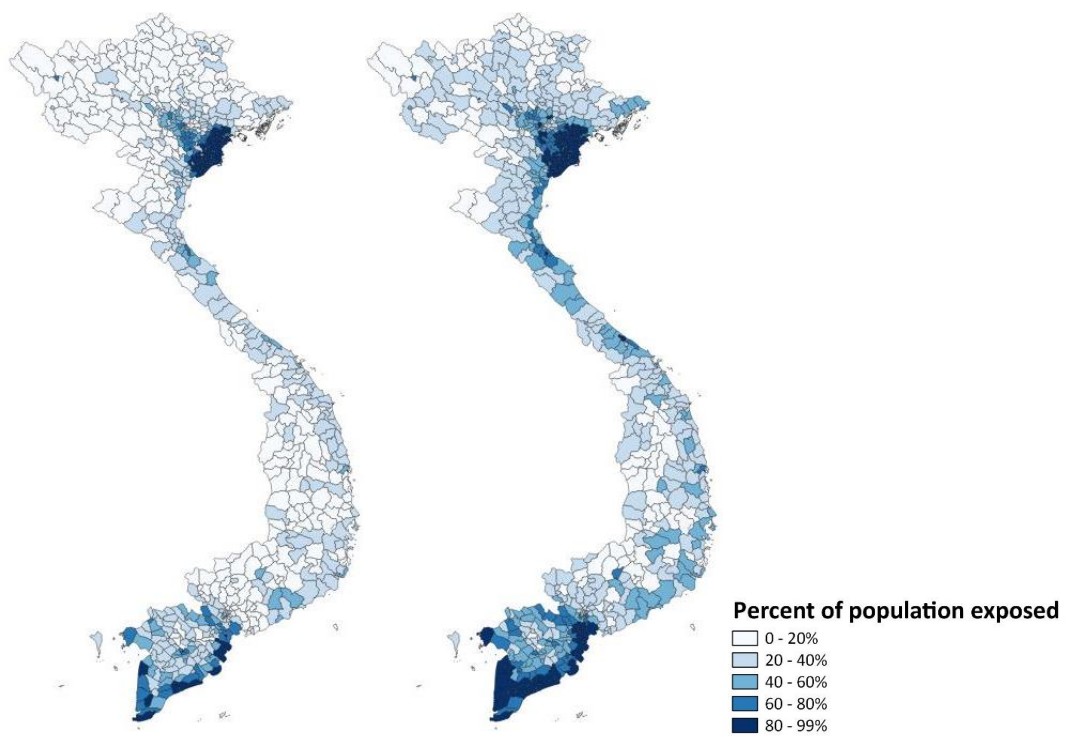

*Map 9. Relative exposure at the district level (% of district population exposed), for a 25-year historical flood (left) and a 25-year flood under high climate change (right).*

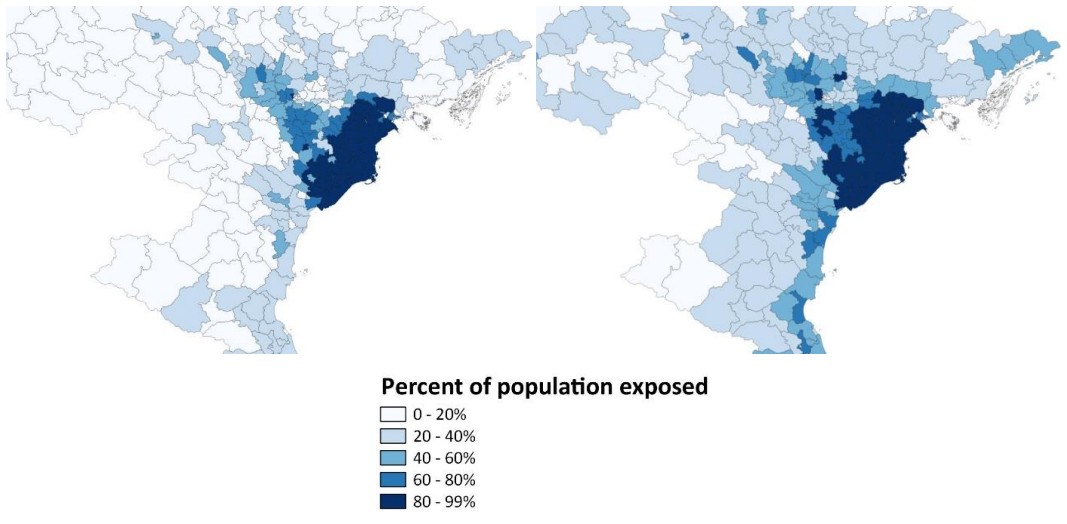

*Map 10. Relative exposure in the Red River Delta for historical 25-year flood (left) and 25-year flood with high climate impacts (right)*





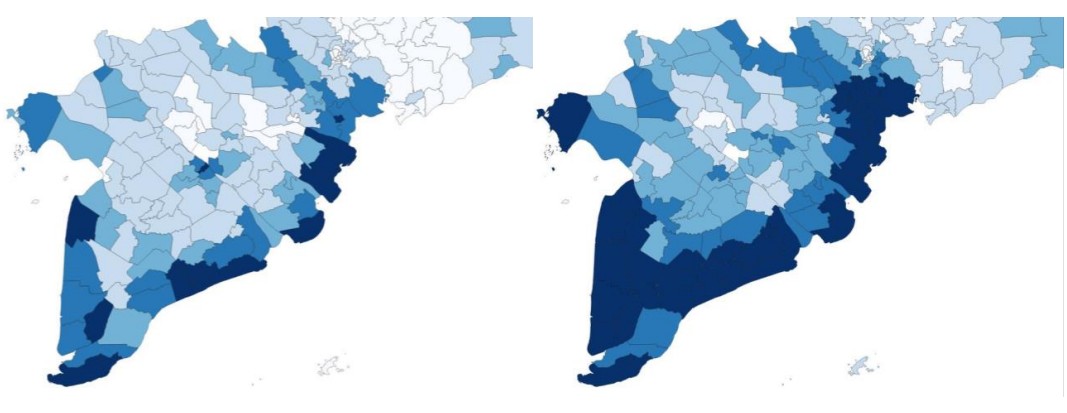

**Percent of population exposed**
☐ 0 - 20%
☐ 20 - 40%
☐ 40 - 60%
☐ 60 - 80%
☐ 80 - 99%

*Map 11. Relative exposure in the Mekong Delta for historical 25-year flood (left) and 25-year flood with high climate impacts (right).*

**5.1.2.**    **Flood exposure and poverty**

Another question is how many poor people are exposed to flood risk in the country. This is important since case studies of poverty and disasters suggests that poor people are more vulnerable to floods (e.g. they lose larger portions of their incomes and assets) and they have less access to support to cope and adapt (Hallegatte et al. 2016).

Livelihood shocks triggered by floods could keep people from escaping poverty and even push them into deeper

poverty (Karim and Noy 2014). Qualitative work undertaken in the provinces of An Giang, Kien Giang, Kon Tum, Hoa Binh and Bac Ninh confirm that many poor households feel more vulnerable to floods due to their increased exposure (a result of living in flood prone areas, like along river banks or outside of protective dikes, and often having substandard quality of housing) and are less likely to have sufficient assets to buffer the effects of floods (World Bank, 2016). Poor households in these provinces also report receiving inadequate support for coping with the aftermath of

floods, and that floods can be one factor in pushing near-poor people into poverty if there is not sufficient safety-net and livelihood support to flood victims (World Bank, 2016).

To examine the question of how many poor people in Vietnam are exposed to flooding, we multiply the population exposure estimates by the district's poverty headcount rate (the percentage of people living below $ USD 1.25 per day), as calculated in (Lanjouw, Marra, and Nguyen 2013).

For a 25-year historical flood, 30% of today's poor population is exposed. This number increases by between 16-28% given climate change impacts. For a 50-year return period under a high climate scenario, 40% of today's poor people in Vietnam are exposed to flooding. For a 200-year return period under a high climate scenario, more than half of



today's poor are exposed. Similar to the population analysis, the impact of climate change on the number of poor people exposed is evident. For instance, a 25-year event with high climate change impacts has the same exposure as a 200-year historical event (at around 41% of poor people being exposed).

*Table 4. Number and percentage of poor exposed to flood risk in Vietnam, across the 16 flood hazard scenarios examined.*

| Scenario | Exposure | Return period | | | |
|---|---|---|---|---|---|
| | | 25 | 50 | 100 | 200 |
| Historical | Estimated poor exposed (million) | 5.28 | 6.19 | 6.88 | 7.24 |
| | Percentage of today's poor | 30% | 35% | 39% | 41% |
| Low climate change | Estimated poor exposed (million) | 6.27 | 6.64 | 7.32 | 7.54 |
| | Percentage of today's poor | 35% | 37% | 41% | 42% |
| | Increase due to climate change | 16% | 7% | 6% | 4% |
| Medium climate change | Estimated poor exposed (million) | 6.80 | 7.16 | 7.69 | 8.00 |
| | Percentage of today's poor | 38% | 40% | 43% | 45% |
| | Increase due to climate change | 22% | 14% | 11% | 10% |
| High climate change | Estimated poor exposed (million) | 7.33 | 7.66 | 8.14 | 8.56 |
| | Percentage of today's poor | 41% | 43% | 46% | 48% |
| | Increase due to climate change | 28% | 19% | 16% | 15% |

Based on the statistics provided in Table 4, there is no strong signal that poor people are more exposed than non-poor people, at the national level. However, this may not be the case in specific regions or within specific districts.

To examine which districts have a confluence of poverty and flood risk, we classify both each district's poverty headcount rate and flood exposure into three categories: low, medium, and high. We create 3 quantiles for each. We examine both absolute and relative numbers, overlaying the number of poor and number of flood exposed, and the percentage of poor and percentage of flood exposed.

The results suggest that areas of the Northern Mountains and the Mekong Delta exhibit districts with high flood and high poverty (darkest shade of brown in Map 12). The results are slightly different when comparing relative and absolute numbers. When using absolute (the number of poor and number of flood exposed) more areas of high flood and poverty are visible in the Mekong and Red River Delta, as well as along the eastern coasts.





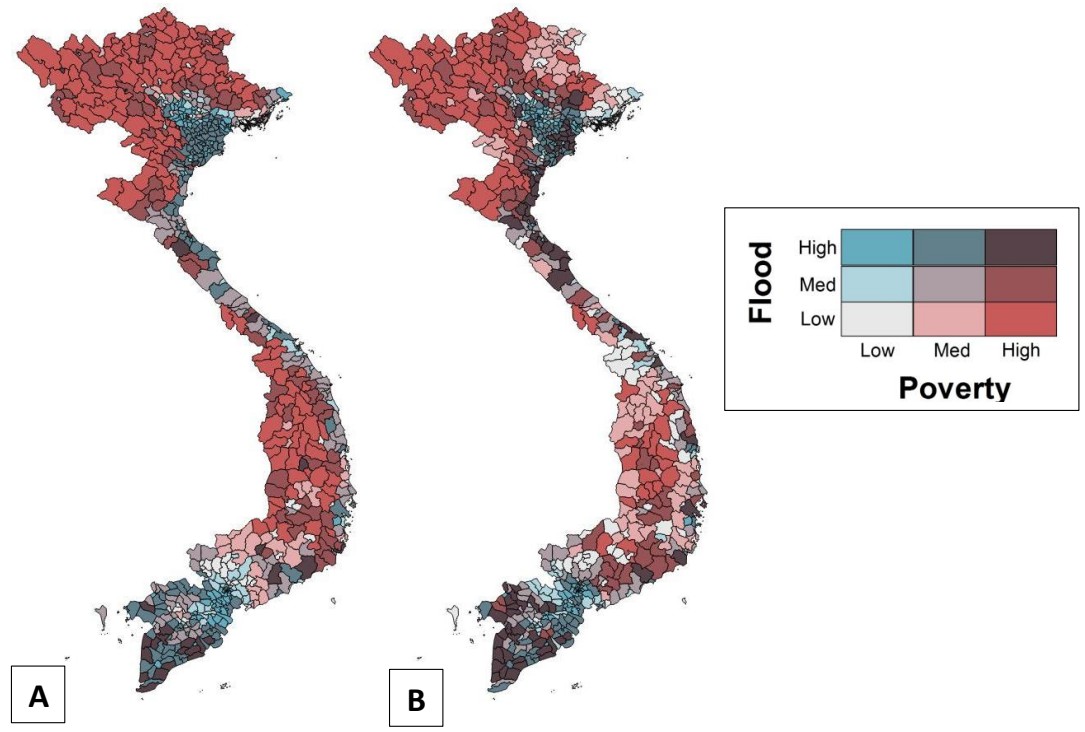

*Map 12. Overlay of poverty and flood at the district level for the 25 year-return period flood with climate change. Map A shows relative exposure, overlaying the % of poor and % of population flooded, Map B shows the absolute exposure, overlaying the # of poor and # of population flooded.*

*Bins: Map A, Poor, Relative (Low = 0-15%, Med = 15-28%, High = 28%+)*
*Bins: Map A, Flood Exposure, Relative (Low = 0-26%, Med =26-47% , High =47%+)*
*Bins: Map B, Poor, Absolute (Low = 0-15,900, Med =15,900 – 31,000, High = 31,000+)*
*Bins: Map B, Flood Exposure, Absolute (Low = 0-27,000, Med =27,000 – 70,000, High = 70,000+)*

However, even though not all of the poorest districts do not seem to face higher exposure risk to floods, it is important to remember that poor households and poor individuals within high exposure areas have generally higher vulnerability to the impact of floods. Further, it is very likely that within a district or city, the poorest are the most exposed to floods.

5    We explore this dynamic at the local scale with a city-level analysis of Ho Chi Minh City.

### 5.2. City-level analysis in HCMC for poverty and exposure to floods

While the relationship between poverty and exposure to floods may not be evident at the national or district level, at a more local scale and especially in urban areas, land and housing markets often push poorer people to settle in riskier areas (Lall and Deichmann 2012). For instance, comparing exposure of poor people to average exposure, poor

10    households are 71% more exposed to flooding in the Mithi River Basin in Mumbai, India (Hallegatte et al. 2016).



We examine these dynamics in Ho Chi Minh City, using high-resolution local-scale flood maps designed specifically for HCMC (Lasage et al. 2014) and proxy for poverty using the spatial location of potential slums from the Platform for Urban Management and Analysis (PUMA) data set (World Bank 2015). The PUMA data set also has information on locations of urban expansion from 2000 to 2012. We therefore examine exposure to flooding in all three locations
– urban areas as a whole, potential slum locations, and areas of urban expansion. The results we present below are for all districts in HCMC; results for only districts with slum areas are similar and thus not reported.

We find that a relatively high percentage of the potential slum areas are exposed to floods, ranging from 68.9% (for a 10-year return period) up to 83.3% (for a 1000-year return period). When considering all urban areas of HCMC, exposure to flooding is lower: 63% (for a 10-year return period) up to 68.3% (for a 1000-year return period). A sea
level rise of 30 cm increases the extent of flooded areas the most in slum areas and for a low-probability but recurrent flood (10-year flood). For a 10-year flood and looking only within slum areas, we find an increase in exposure of 15 percentage-points due to sea level rise, compared to a difference of 5.7 percentage-points when looking at the entire urban area of HCMC. These results, as presented in Figure 1, suggest slum areas to be more exposed to floods than non-slum areas.[4]

Due to cognitive biases, it can be hypothesized that flood risk from frequent events (like the 10-year return period event) are more likely to be remembered than a rare event (like the 1000-year return period event) and thus more likely to be included in land values. If this logic holds, it is likely that potential slum areas should exhibit a higher exposure than other areas for frequent events. However, in our analysis we find the opposite: that the difference between slum and non-slum increases as the return period event gets rarer.

When looking at the areas of urban expansion we find that a large share, 72.2% under a flooding event with a 10-year year return period up to 74.4% in case of a 1000-year return period, of these areas is located in areas prone to flooding (Figure 1).

---

[4] Disaggregated results per district can be found in Appendix 2. Results using depth as an indicator for flooding is also presented in Appendix 2.




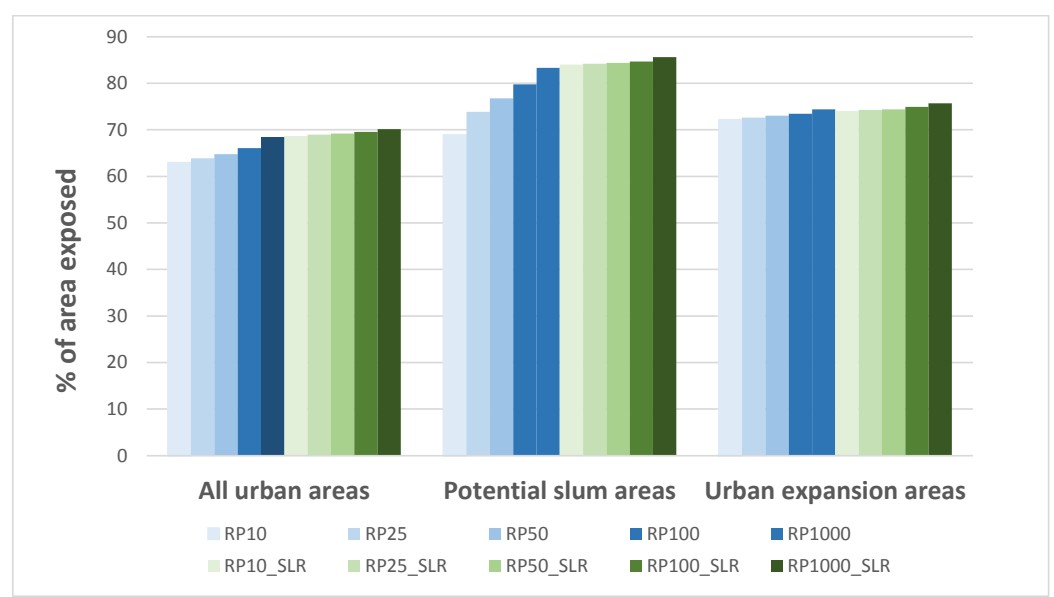

*Figure 1. Slum areas tend to be more exposed than the average, across all flood scenarios. SLR means the scenario includes a 30cm sea level rise due to climate change. RP denotes the turn period event.*

### 6. Limitations

The findings presented should be interpreted considering a number of caveats.

While we use current and future flood data, we only use current population and poverty data, as reliably projecting these socioeconomic trends spatially into the future is almost impossible. Changes in these trends – among many other factors – can lower socioeconomic vulnerability even as the climate change hazard increases (Hallegatte et al. 2016). Along these lines, while we examine which regions within Vietnam have the highest flood exposure, we do not

examine the determinants of vulnerability (other than poverty). Recent analyses suggest that the Northwest, Central Highlands, and Mekong River Delta have the greatest socioeconomic vulnerability (World Bank 2010).

In the flood hazard maps developed for this paper, we assume no protection due to a lack of data and as a result the hazard maps present an upper bound of flood risk. Work is currently ongoing to develop a global database of flood protection, and this information can be mobilized for future work (Scussolini et al. 2015). For the national-level

analysis, flooded areas are defined as any area with inundation higher than 0. We have not yet explored the depth dimension, although the flood hazard maps developed for this study allow for this potential in future work.

For the HCMC analysis, the location of the slum areas in the PUMA data set are mainly restricted to the old town. Furthermore, slum areas are often difficult to define and the data we have likely does not capture all slum areas within HCMC. Finally, the urban expansion data set does not make a distinction between the urban expansion of residential

areas or infrastructure (roads, etc.).





In terms of the hazard, the flood maps for HCMC show flood depth and extent from the river and from sea (when looking at the sea level rise scenario). Pluvial flooding and possible 'sink'-areas in the city are not taken into account. Moreover, the lowest return period we have flood maps for is not low, compared to what is experienced in the city. Some areas of HCMC are flooded every year. Since this analysis used a flood with a 10-year return period as the flooding scenario with the highest recurrence interval we were not able to capture the relative differences in exposure to these yearly/bi-annual flooding events (and we hypothesize that poor people are relatively more exposed to these types of flooding than the general population).

## 7. Conclusions

Despite the limitations, this analysis presents some initial findings on what exposure to floods looks like in Vietnam, how it may change under a changing climate, and the exposure of poor people. The results from this paper have implications regarding infrastructure development, land use planning, and strategies to manage flood risk.

First, the results of this paper suggest that climate change is likely to increase the number of people exposed to floods, especially in the Mekong and Red River Deltas. Climate change impacts can make frequent events as important as rare ones in terms of exposure: for instance, a 25-year flood under future conditions of high sea level rise exposes more people than a 200-year flood under current conditions. In addition to showing the benefits of investing in flood risk management today, these results also suggest that new investments in flood protection (whether natural protection through mangroves or physical protection through dikes and drainage systems) should be planned with climate change considerations (e.g. where will flood be the worst in the future and how can new infrastructure withstand future levels of flood).

However, current planning approaches in Vietnam have not yet adequately taken these existing and future floods into account (IMHEN and UNDP 2015). For example, the city of Long Xuyen in the Mekong Delta has based its dike infrastructure around the city on historical floods levels only, with no inclusion of future climate change-induced water levels (World Bank, 2016). Qualitative surveys in Long Xuyen suggest existing defenses have already proved inadequate in recent flooding (World Bank, 2016). Investments in climate-informed flood protection taken now reduce flood exposure, but can also save money in the long-run by reducing the amount spent on recovery and reconstruction for future floods. And while it is challenging to integrate into project planning, innovative approaches such as decision-making under uncertainty can provide support to decision-makers on how to design flood projection with climate change in mind (Hallegatte et al. 2012).

Furthermore, while infrastructure investment can protect certain areas, it may increase exposure in other areas. for example, upstream dams in the Mekong Delta can increase the strength and velocity of downstream floods (World Bank, 2016). Thus, flood risk assessment needs to be integrated across policy sectors (agriculture, industry, infrastructure, defense, urban planning) before development decisions are made, and more comprehensive approaches, like integrated coastal zone management and planning, would help to reduce exposure of people and assets in Vietnam.



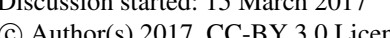


Second, this paper found that that while poor people do not appear to be more exposed to floods than non-poor people at the national level, this is not the case at the local level within a city. The results presented suggest potential slum areas are more exposed to floods than non-slum areas, and the exposure differential increases with sea level rise. As a result, risk-sensitive land-use planning may be a priority to ensure development takes place in safer areas.

In many cases, risk-sensitive land-use planning involves resettlement, which is the major ex-ante hazard adaptation mechanism employed in Vietnam currently, especially in the Mekong Delta. Furthermore, recent research in Tan Chau district suggests the resettlement policy enacted in 2002 may have made households worse-off. Inadequate financing resulted in households paying for their new settlements out-of-pocket; many households who were farmers and fishers did not have adequate land, transportation and market access, and inadequate livelihood support was provided to them
(World Bank, 2016). Where resettlement policies are enacted, it is imperative that such policies are paired with livelihood and financing support.

The estimates of increasing exposure provided in this paper also provide support for increased attention towards strategies which reduce vulnerability or increase the ability of households to adapt to floods. Strategies such as government subsidies for household-level flood protection (like raising of floors), improved financial inclusion, and
better observation systems and early warning, and resilient agricultural practices can reduce the asset and income losses associated with floods (Hallegatte et al. 2016). And when hit, targeted social protection (which can support the affected population quickly after a large flood) can hasten recovery (Hallegatte et al. 2016, Chapter 5). Such policy measures may be targeted in areas with higher future exposure (geographical targeting) as well as to individuals and households classified as poor and near poor who experience flooding (individual targeting). Areas such as the Northern
Mountains have high poverty and are expected to experience an increase in flood exposure. While infrastructure protection can be costly in these remote and sparsely-populated areas, strategies to reduce vulnerability or improve the ability-to-adapt of households can reduce flood impacts.

Risk management policies are best designed as holistic strategies that combine many of these levers – from risk-sensitive land-use planning to flood protection to investments in social protection (World Bank 2013). But each
strategy undertaken will be context-specific and based on local conditions. The results of this paper provide some insights to inform the implementation of risk management policies in Vietnam and suggest that such investments can better manage current and future flood risks if action is taken today.






## 8. Code availability

The code is freely available upon request.

## 9. Data availability

The data is freely available upon request.

## 10. Appendices

### 10.1. Appendix 1. National-level: Flood Hazard Modeling Details

All the data were produced using the SSBN global flood model, producing flood hazard data at 90m resolution. The SSBN global model couples a flood frequency analysis conducted at the global scale, with a fully 2-D hydraulic modeling framework. Extreme river discharges are derived from a Flood Frequency Analysis (FFA), applied at the

global scale (Smith, Sampson, and Bates 2015). The model also explicitly simulates in-channel flow, with the FFA also used to estimate bankfull discharge across the channel network (defined as the $1.1 - 2$ year event depending on climate zone); these values are used to calibrate the channel conveyance capacity within the hydraulic modeling framework. A number of global data sets are used to derive the inputs to the hydraulic model. Firstly, the Hydrosheds variant of SRTM is used, both at 3 and 30 arc second resolutions. A number of additional corrections are applied to

the terrain data including a systematic vegetation correction procedure in vegetated areas and an urban correction procedure in urbanized areas. A detailed description of the modeling framework is provided by (Sampson et al. 2015).

For the coastal simulations, input boundary conditions were derived using estimates of return period surge heights, taken from (Chinh, JianCheng, and Bui 2014). Storm surge hydrographs for each recurrence interval were taken for four tidal gauges located along the Vietnamese coastline. Coastal boundary conditions for the hydraulic model were

derived by linearly interpolating between the gauge locations. The hydraulic model was set up so that a coastal boundary condition was implemented for each 'land' cell located next to the coast. In addition to the coastal boundary conditions, large river channels were also included in the simulations, using a sub-grid channel network set-up (Neal, Schumann, and Bates 2012). In channel flow was estimated to be 0.5*bankfull Q; rivers were estimated to be at 50% channel capacity.

Coastal simulations under future climate conditions were undertaken using the latest projections of global mean sea-level rise, outlined in the Fourth Assessment Report (AR5) and Fifth Assessment Report (AR5) of the Intergovernmental Panel on Climate Change (IPCC) (IPCC 2014; IPCC 2007). Estimates of sea-level changes were taken and used to directly perturb the boundary conditions used in the simulations under current conditions. In order to incorporate uncertainty, simulations were undertaken for a range of projected changes, represented here as Low,

Medium and High sea-level rise (SLR) projections (

Table 1). The simulations are all conducted assuming that no flood defenses are in place; clearly flood defenses are not represented in the available terrain data. Therefore, these simulations should be considered as an upper bound of flood risk in the country.





The possibility of including a storm surge intensity component to the future projections was also discussed, but there are significant uncertainties around quantifying storm intensity changes that would preclude any reasonable modeling being undertaken (Seneviratne et al. 2012). It also seems that changes in surge extremes are going to be largely driven by sea-level rise (Lowe et al. 2010). That being said, there are studies attempting to quantify changes in storm surge

5     intensity; Lin et al. (2012) reported in a study focused on the North Atlantic, that in some cases changing storm surge intensity was comparable to changing sea-level rise. Such changes would effectively double change in hazard intensities presented here. As of yet, we have not included this in the simulations due to the uncertainty.

### 10.2.     Appendix 2. City-case: HCMC

#### 10.2.1.     Results for flood extent across districts

10     Figure 2 presents the differences in relative flooding extent (%) for a flooding event with a 10 year or 50 year return period representing the current and future conditions. Overall, we find that both under current conditions and given a 30 cm sea level rise, relative flooding extent is higher in slum and urban expansion areas than in the non-slum areas, both when looking at the HCMC-city level and at the level of districts. This can serve as a first-order indicator that these slum and urban expansion areas are relatively often located in flood prone areas.

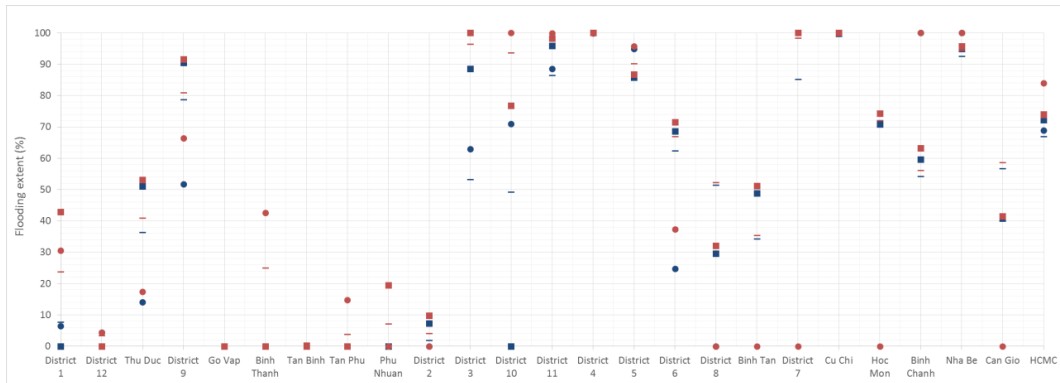

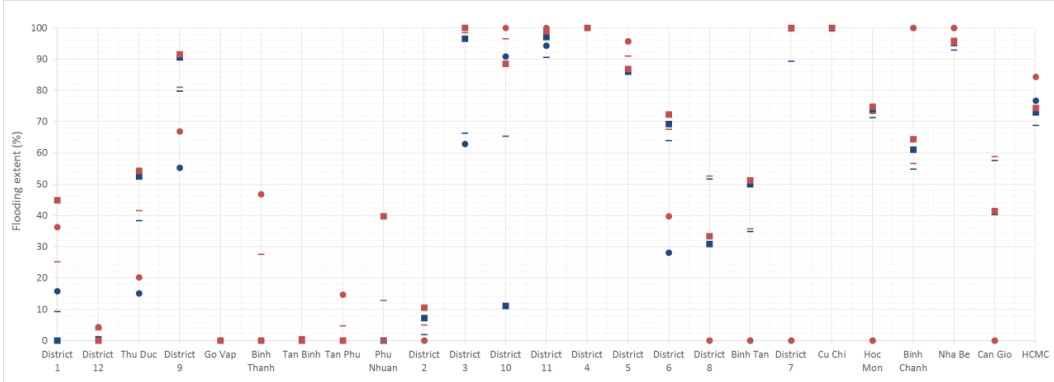





■ Sprawl areas - Current conditions

— Non-slum areas - Current conditions

● Slum areas - Future conditions

■ Sprawl areas - Future conditions

— Non-slum areas - Future conditions

*Figure 2. Graph showing the differences in relative flooding extent (from 0 to 100%) for a (a) 1/10 and (b) 1/50 year flooding event under current and future conditions per district and for the whole city of HCMC between all urban non-slum areas, slum areas and areas of urban expansion.*

### 10.2.2. Flood depth analysis

We also examine the exposure to flooding in terms of mean flooding depth. At the city-level, mean inundation depths were found to be higher in the urban non-slum area compared to the slum locations under any of the return periods used (Figure 3). However, spread (standard deviation) in inundation depths was found to be very large when looking at the city-totals. Looking at the individual districts, we find a higher inundation depth within slums – compared to non-slum areas – in five districts for the 10-year return period flood up to eight districts for the 1000-year return period flood. A sea-level rise of 30cm increased mean inundation depths for the entire city by 30–40 cm depending on the return period. No significant differences were found in the increase in inundation depths between slum and non-slum areas. When looking at the areas of urban expansion, we found that mean inundation depths –irrespective of the choice of return period - are higher in these areas compared to the general non-slum urban area of HCMC, both when looking at the total city of HCMC and in a majority of districts.

Mean inundation depths are found to be higher in the areas where urban expansion takes place compared to the general urban area of HCMC, both when looking at the total city of HCMC and in a majority of districts.

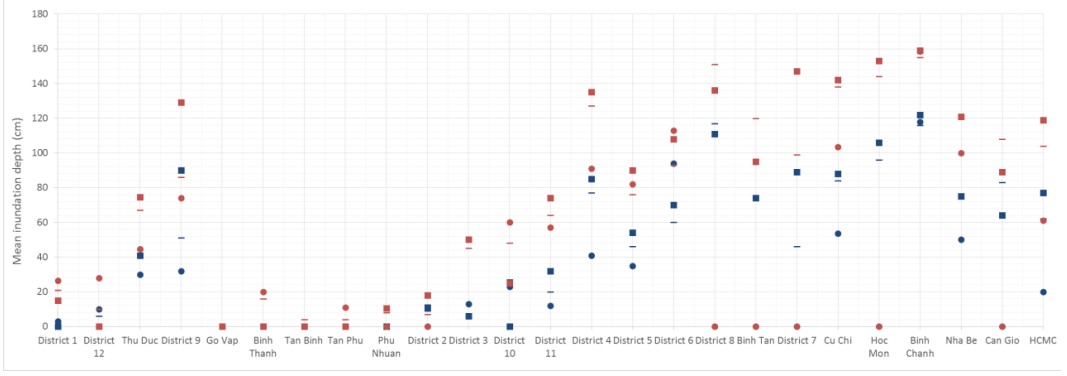





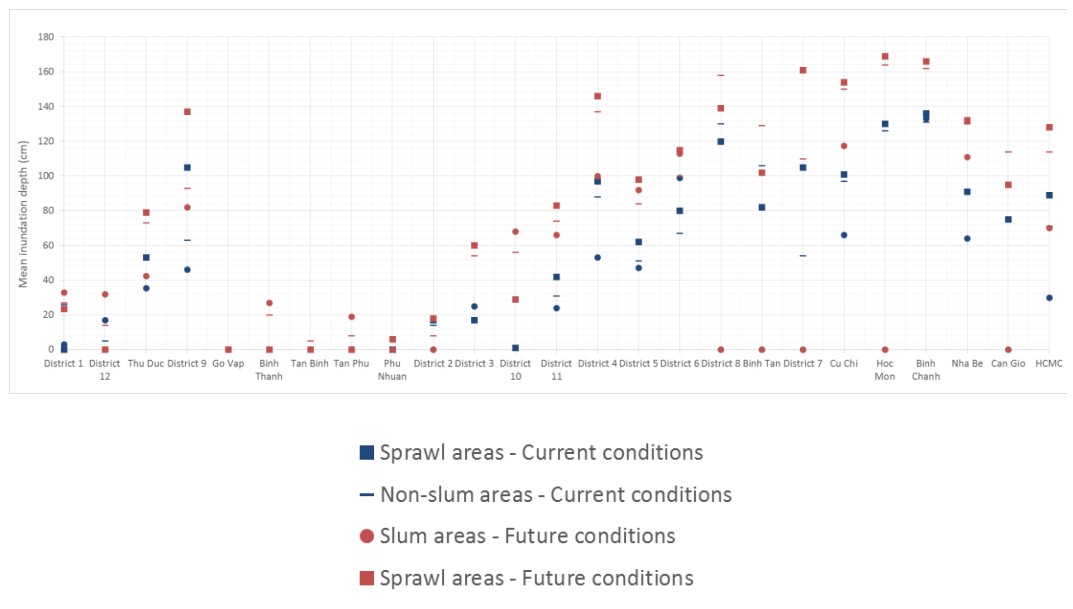

*Figure 3. Graph showing the difference in mean inundation depth (from 0 to 180cm) between slum areas, locations with urban expansion and urban non-slum areas, per district and for the whole city of HCMC using a flooding scenario with a (a) 1/10 and (b) 1/50 year return period representing current and future conditions.*

## 11. Author contribution

Andrew Smith conducted the hydrological modeling of the national-level flood hazard maps. Ted Veldkamp produced the flood maps and conducted the analysis of Ho Chi Minh City. Mook Bangalore designed and executed the research, and also prepared the manuscript, with contribution from all co-authors.

## 12. Acknowledgements

The authors declare that they have no conflict of interest. This work is part of the programmatic AAA on Vietnam Climate Resilience and Green Growth (P148188) and was developed under the oversight of Christophe Crepin at the World Bank. It contributed to the global program on Climate Change and Poverty (P149919) under the oversight of Stephane Hallegatte at the World Bank. The authors thank Abigail Baca, Christophe Crepin, Chandan Deuskar, Stephane Hallegatte, Stuart Hamilton, Pam McElwee, Madhu Raghunath, Maurice Rawlins, Ulf Narloch, Dzung Huy Nguyen, and Vo Quc Tuan for valuable comments and feedback. The authors may be contacted at M.Bangalore@lse.ac.uk

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
