# Peer review of "Exposure to Floods, Climate Change, and Poverty in Vietnam"

_Natural Hazards and Earth System Sciences, 2017_

## Referee Comment (RC1) · Anonymous Referee #1 · 18 Apr 2017

The paper deals with a very interesting topic. It shows the exposure on the one hand of the Vietnamese population and on the other hand of the population in HCMC to current and future flooding. Interestingly, the authors find out that at the local level especially the poor are highly exposed. The paper has a clear structure and is well written. All in all, I think that this paper has the potential to be a major contribution. Overall, I recommend a revision.

I restrict my comments to the following aspects:

1. The paper is a pure empirical paper on the exposure. As there are many studies on the exposure to risks in Vietnam, the question is really about the value added of this contribution. In my opinion a strong selling point is the consideration of socioeconomic characteristics and the focus on the poor. Therefore, a stronger theoretical foundation

on the relationship between poverty and vulnerability is suggested.

2. The authors promise that "we combine high-resolution flood hazard data with spatial data on slum location, urban expansion, and migration...". In my opinion, nothing is really said on urban expansion, and migration. It would be very good to discuss development trends of HCMC – to get a much better understanding of the context conditions, and the driving forces behind the urban expansion.

3. The reader is more or less left alone with the results. The authors claim to have shown the "benefits of investing in risk and flood management today". But, no in-depth discussion on the possible positive advantages of introducing a management system has been addressed. The central question is how such a management system should look like and how it could be implemented in HCMC or Vietnam – where institutional framework conditions are not easy.

---

## Referee Comment (RC2) · Anonymous Referee #2 · 5 Jul 2017

In the manuscript, authors assess the distribution of flood exposure among poor and non-poor locations. The topic is interesting and broad. However, the manuscript is far away from publication. I would suggest 'major revision' taking into account following comments:

1. Authors have used the term, 'hazards' and 'risk' interchangeably throughout the manuscript. Authors should take necessary care and consistency of using these terminologies.

2. The paper seems much more policy oriented than academic journal article. How the research contributes to the field of natural hazards is not clear. Certainly the research support to Vietnamese policy makers, but what are the benefits of general readers of the natural hazards (science) community is a big question mark.

[Figure]

3. Instead of only considering Vietnamese context, authors should provide state of art on the issue and they should explicitly consider some innovativeness in their research. In first two paragraphs of literature review section, it seems that this kind of research has already been done elsewhere. Therefore, why another similar research is required to the science community? I would suggest to revisit the manuscript.

4. As 'literature review' is usually considered in academic thesis paper (not for journal article), I would suggest to include them within introduction section for better representing state-of-art.

5. I would suggest to consider some recent articles on flood risks in Vietnamese context (e.g., Apel et al., 2016; Chinh et al., 2016, 2017).

6. Authors have considered 'head count rate' for assessing poverty. There are also other indices for assessing poverty. Authors should provide a justification of their choice.

7. In table 3, 'm' within bracket: does it denotes millions? Authors should explicitly define this.

8. No validation of the simulated results has been done except footnote 3 (on Jongman et al. 2014). Is there any national statistics on historical flood exposed population?

9. In the conclusion, I would suggest to generalize some results from the analysis that can also be useful for other areas.

Suggested References Apel, H., MartínezÂăTrepat, O., Hung, N. N., Chinh, D. T., Merz, B., and Dung, N. V.: Combined fluvial and pluvial urban flood hazard analysis: concept development and application to Can Tho city, Mekong Delta, Vietnam, Nat. Hazards Earth Syst. Sci., 16, 941-961, https://doi.org/10.5194/nhess-16-941-2016, 2016.

Chinh, D.T.; Gain, A.K.; Dung, N.V.; Haase, D.; Kreibich, H. Multi-Variate Analyses of Flood Loss in Can Tho City, Mekong Delta. Water 2016, 8, 6.

[Figure]

Chinh, D.T.; Dung, N.V.; Gain, A.K.; Kreibich, H. Flood Loss Models and Risk Analysis for Private Households in Can Tho City, Vietnam. Water 2017, 9, 313.

---

## Author Comment (AC1) · 14 Aug 2017

The paper deals with a very interesting topic. It shows the exposure on the one hand of the Vietnamese population and on the other hand of the population in HCMC to current and future flooding. Interestingly, the authors find out that at the local level especially the poor are highly exposed. The paper has a clear structure and is well written. All in all, I think that this paper has the potential to be a major contribution. Overall, I recommend a revision.

I restrict my comments to the following aspects:

1. The paper is a pure empirical paper on the exposure. As there are many studies on the exposure to risks in Vietnam, the question is really about the value added of this contribution. In my opinion a strong selling point is the consideration of socioeconomic characteristics and the focus on the poor. Therefore, a stronger theoretical foundation on the relationship between poverty and vulnerability is suggested.

Thanks for your comment, and overall review. Yes, we also agree that the value added of this contribution pertains to the focus on socioeconomic characteristics, and poverty in particular. One of the co-authors recently co-authored a report on poverty and disasters (https://openknowledge.worldbank.org/handle/10986/25335), so the theoretical foundation from that report can be adapted to the Vietnam context and added to the paper.

2. The authors promise that "we combine high-resolution flood hazard data with spatial data on slum location, urban expansion, and migration. . .". In my opinion, nothing is really said on urban expansion, and migration. It would be very good to discuss development trends of HCMC – to get a much better understanding of the context conditions, and the driving forces behind the urban expansion.

We will revise this statement.  We have used the PUMA dataset to look at urban expansion and expansion/location of slums over the period 2000-2012. We don't discuss migration nor do we have any information on the future development trends in HCMC. To accommodate your comment we will put the PUMA datasets a bit more in perspective and investigate literature on population growth/ urban growth in HCMC over the period 2000 – 2012 to back up the PUMA data.

3. The reader is more or less left alone with the results. The authors claim to have shown the "benefits of investing in risk and flood management today". But, no in-depth discussion on the possible positive advantages of introducing a management system has been addressed. The central question is how such a management system should look like and how it could be implemented in HCMC or Vietnam – where institutional framework conditions are not easy.

Thanks for your comment. This is indeed a complex question, but our results can provide some insights on where geographically infrastructure to reduce flooding might take place. Generally these decisions are based upon which areas have the highest property and asset values, but do not consider where poor people live. Poor people don't own much, so this won't be reflected in asset values, but they suffer the

most. As a result, we can offer some suggestions of how to incorporate these socio-economic characteristics when planning where to invest.

At the urban scale, the issue is very different. HCMC is continuously growing, and one possible implication of the finding that slums are highly exposed to floods is to try to encourage development in safer parts of the city, linked to the city center by transport. We can identify such areas in the paper. However, the institutional challenge remains and becomes even more binding, as such a strategy would require coordination between many agencies – flood management, urban planning, and transport.

---

## Author Comment (AC2) · 14 Aug 2017

In the manuscript, authors assess the distribution of flood exposure among poor and non-poor locations. The topic is interesting and broad. However, the manuscript is far away from publication. I would suggest 'major revision' taking into account following comments:

1. Authors have used the term, 'hazards' and 'risk' interchangeably throughout the manuscript. Authors should take necessary care and consistency of using these terminologies.

We will revise. Thanks for pointing this out.

2. The paper seems much more policy oriented than academic journal article. How the research contributes to the field of natural hazards is not clear. Certainly the research support to Vietnamese policy makers, but what are the benefits of general readers of the natural hazards (science) community is a big question mark.

While this research does support policy makers in the region, we feel as if it also contributes to the academic research on natural hazards. This paper is the first to combine high-resolution flood hazard modeling with spatially-explicit poverty maps. In addition, we assess how flood exposure is distributed across poor and non-poor areas, at the country and city levels. We believe the combination of these datasets – typically kept separate in the field – and the analysis at multiple levels is a contribution to the science community.

3. Instead of only considering Vietnamese context, authors should provide state of art on the issue and they should explicitly consider some innovativeness in their research. In first two paragraphs of literature review section, it seems that this kind of research has already been done elsewhere. Therefore, why another similar research is required to the science community? I would suggest to revisit the manuscript.

When examining flood exposure, it is important to get as local as possible since impacts can vary widely across space. Indeed, unlike other perils flooding is spatially complex, requiring local features to be resolved sufficiently in order to obtain an accurate picture of flood risk. While previous studies mostly focus on dynamics at the country level, the contribution regarding state-of-the-art is the high resolution at which the analysis is conducted. At best, the studies listed in the literature review use hazard data that are resolved at a resolution of ~1km². The flood hazard model used here represents a true state-of-the-art model, using a fully 2D hydrodynamic model to resolve flood hazard at a resolution of ~90m². Furthermore, the poverty maps used are at the district level, which provide a precise estimate of poverty status across the country.

In addition, in line with another reviewer's comment, we will add a more general theoretical underpinning of the topic of flood risk and poverty, which will further clarify the value added of this work.

4. As 'literature review' is usually considered in academic thesis paper (not for journal article), I would suggest to include them within introduction section for better representing state-of-art.

Thanks for your suggestion. We will review other state-of-the-art studies and adjust the structure accordingly.

5. I would suggest to consider some recent articles on flood risks in Vietnamese context (e.g., Apel et al., 2016; Chinh et al., 2016, 2017).

Thanks for sharing these. We will take a look and include.

6. Authors have considered 'head count rate' for assessing poverty. There are also other indices for assessing poverty. Authors should provide a justification of their choice.

For Vietnam, the only data available is on the headcount rate and on the headcount. Other indices for poverty were not available at the district-level for Vietnam. We will add this to the paper as a justification for our choice.

7. In table 3, 'm' within bracket: does it denotes millions? Authors should explicitly define this.

Yes, that is millions. Thanks for pointing it out. We will revise.

8. No validation of the simulated results has been done except footnote 3 (on Jongman et al. 2014). Is there any national statistics on historical flood exposed population?

While there are some reports with images, details on national statistics on historical flood exposed population are unavailable. However, when we discussed this paper with colleagues on the ground in Vietnam, they tended to agree that the numbers were in the range of estimates they expected. We can also check our results against past disaster events in Vietnam, using the EM-DAT database on disasters.

9. In the conclusion, I would suggest to generalize some results from the analysis that can also be useful for other areas.

Thanks for this comment. We will revise.

Suggested References Apel, H., Martínez‐aTrepat, O., Hung, N. N., Chinh, D. T., Merz, ̌ B., and Dung, N. V.: Combined fluvial and pluvial urban flood hazard analysis: concept development and application to Can Tho city, Mekong Delta, Vietnam, Nat. Hazards Earth Syst. Sci., 16, 941-961, https://doi.org/10.5194/nhess-16-941-2016, 2016.

Chinh, D.T.; Gain, A.K.; Dung, N.V.; Haase, D.; Kreibich, H. Multi-Variate Analyses of Flood Loss in Can Tho City, Mekong Delta. Water 2016, 8, 6.

Chinh, D.T.; Dung, N.V.; Gain, A.K.; Kreibich, H. Flood Loss Models and Risk Analysis for Private Households in Can Tho City, Vietnam. Water 2017, 9, 313.